# PREDICTION TASKS IN GRAPHS: A FRAMEWORK TO CONTROL THE INTERPRETABILITY-PERFORMANCE TRADE-OFF

## ABSTRACT

Graph Neural Networks (GNNs) have emerged as state-of-the-art methods for solving graph-level tasks in diverse domains, such as social network analysis and bioinformatics. However, their complex nature results in a lack of human-interpretable predictions, which can hinder their practical impact. Here, we aim at improving GNN interpretability by targeting *sparsity* during GNN training, i.e., by minimizing the size (and/or number) of subgraphs used to make predictions. Existing solutions in the literature suffer from two main limitations: i) they still rely on information about the entire graph; and/or ii) they do not allow practitioners to directly control the trade-off between predictive performance and sparsity. To address the above limitations, in this paper, we formulate GNN training as a bi-level optimization task, where the trade-off between interpretability and performance can be controlled by a hyperparameter. Our framework relies on reinforcement learning to iteratively maximize predictive performance and sparsity by removing edges or nodes from the input graph. Our empirical results on nine different graph classification datasets show that our method competes in performance with baselines that use information from the whole graph, while relying on significantly sparser subgraphs, leading to more interpretable GNN-based predictions.

## 1 INTRODUCTION

Graph Neural Networks (GNNs) have become a cornerstone in modern machine learning, excelling in diverse domains such as social network analysis Cao et al. (2019b;a), recommender systems Fan et al. (2019); Bai et al. (2020) and bioinformatics Guo et al. (2021); Ramirez et al. (2020); Xiong et al. (2020).However, the complexity of GNNs contributes to one of their principal shortcoming: a lack of human-interpretable predictions. This opacity hinders their potential for practical, real-world impact, as practitioners often require interpretable models to inform decision-making, ensure trustworthiness, and comply with regulations Doshi-Velez & Kim (2017).

Current interpretability methods obtain the most relevant subgraph from an original graph using a GNN-based predictor Sun et al. (2021). In essence, interpretability in this context is linked to graph sparsity. This means using a minimal set of nodes and edges from the graph for prediction rather than the entire graph. A sparser graph reduces the volume of information used in predictions, making it easier for humans to understand. It is crucial to note that sparsity achieves interpretability only when the subgraph completely excludes information from the omitted nodes and edges. This scenario does not occur, for instance, if message passing is employed before obtaining the sparser graph. Some methods aim to identify subgraphs during training Cangea et al. (2018); Sun et al. (2021). However, these methods suffer from two primary shortcomings: i) they typically depend on the entire graph's information for predictions Cangea et al. (2018), which can complicate explanations, and ii) they do not permit direct control of the trade-off between predictive performance and sparsity. In practice, this adaptability is highly desirable. For instance, accuracy might be prioritized in specific applications such as community detection in social networks Contisciani et al. (2020); Shchur & Günnemann (2018). Conversely, in other application scenarios, such as risk assessment Bi et al. (2022), the capability to interpret predictions becomes essential and cannot be traded by model performance.

In this paper, we present a novel framework named GCIP. This framework enables interpretable GNN-based predictions in graphs, and to control the interpretability-performance trade-off. We jointly optimize performance and sparsity via a bi-level optimization strategy. Specifically, sparsity is

achieved through a policy-based reinforcement learning method, which aids in node/edge removal decision-making. We propose a reward function with two parameters that enable the addition of inductive biases toward sparsity or performance. Our framework allows practitioners to specify the type of removal (node or edge) and easily control the sparsity-performance trade-off.

To summarize, the key contributions of our work are: (i) A novel framework to solve graph-level classification using sparse subgraphs for prediction, leading to more interpretable solutions. (ii) The possibility to choose between achieving sparsity through node or edge removal, which broadens the range of application scenarios. (iii) The design of a reward function that allows for a priori inductive biases towards sparser or more high-performing solutions (adaptability). (iv) A comprehensive comparison of nine graph classification datasets, demonstrating that GCIP achieves competitive performance while utilizing significantly sparser graphs for predictive tasks.

## 2 PRELIMINARIES

This section outlines necessary background information on two key building blocks of the proposed framework: reinforcement learning and message-passing graph neural networks. We begin by introducing the notation used throughout this work.

**Notation.** A graph is denoted as $\mathcal{G} = (\mathcal{V}, \mathcal{E})$ where $\mathcal{V} \in 1, \ldots, n$ is the set of $n$ nodes and $\mathcal{E} \subseteq \mathcal{V} \times \mathcal{V}$ is the set of edges. The size of a set is represented as $|\mathcal{V}| = n$. A subgraph of $\mathcal{G}$ is denoted as $\mathcal{G}_s = (\mathcal{V}_s, \mathcal{E}_s) \subseteq \mathcal{G}$ comprising subsets of nodes $\mathcal{V}_s \subseteq \mathcal{V}$ and edges $\mathcal{E}_s \subset \mathcal{E}$. A labeled dataset is represented as $\mathcal{D} = \{\mathcal{G}_i, \boldsymbol{y}_i\}_i$ where $\boldsymbol{y}_i$ denotes the target of $\mathcal{G}_i$. Here, without loss of generality, we focus on $K$ classes graph classification, hence $\boldsymbol{y}_i \in 1, \ldots, K$.

### 2.1 REINFORCEMENT LEARNING

Reinforcement Learning (RL) is a learning paradigm in which an agent learns to optimize decisions by interacting with an environment Sutton & Barto (2005). The objective is to derive a policy $\pi$ that maximizes the expected cumulative reward. An RL problem is typically modeled as a Markov Decision Process (MDP), defined by a tuple $(\mathcal{S}, \mathcal{A}, \mathcal{P}, \mathrm{R}, \gamma)$, where $\mathcal{S}$ and $\mathcal{A}$ denote the state and action spaces, $\mathcal{P}$ the transition probability, $\mathrm{R}$ the reward function, and $\gamma$ the discount factor.

**Policy Gradient Methods.** Policy gradient methods directly optimize the policy using gradient ascent on the expected cumulative reward Sutton et al. (1999). Proximal Policy Optimization (PPO) Schulman et al. (2017b) is a policy gradient method that introduces an objective function fostering exploration while mitigating drastic policy updates. PPO aims to solve the optimization problem:

$$L^{CLIP}(\phi) = \mathbb{E}_t \left[ \min \left( r_t(\phi) \hat{A}_t, clip(r_t(\phi), 1 - \epsilon, 1 + \epsilon) \hat{A}_t \right) \right] \tag{1}$$

where $r_t(\phi) = \frac{\pi_\phi(a_t|s_t)}{\pi_{\phi_{old}}(a_t|s_t)}$ is the probability ratio, $\hat{A}_t$ an estimator of the advantage function at time $t$, and $\epsilon$ a hyperparameter controlling the deviation from the old policy.

### 2.2 GRAPH NEURAL NETWORKS

Message passing Graph Neural Networks (GNNs) are designed for graph-structured data processing. Each node $v \in \mathcal{V}$ has a feature vector $\boldsymbol{h}$, and the GNN transforms these features into more informative representations using neighborhood information. Formally, a GNN performs $K$ update rounds, updating each node's feature vector $\boldsymbol{h}_i^{(k)}$ at step $k \geq 1$:

$$\boldsymbol{h}_i^{(k)} = \mathrm{f}_{\theta_u} \left( \bigoplus_{j \in \mathcal{N}_i} \mathrm{m}_{\theta_m}(\boldsymbol{h}_i^{(k-1)}, \boldsymbol{h}_j^{(k-1)}) \right). \tag{2}$$

Here, $\mathrm{f}_{\theta_u}$ and $\mathrm{m}_{\theta_m}$ are differentiable, parameterized functions, $\mathcal{N}_i$ represents node $i$'s neighbors, and $\bigoplus$ denotes permutation invariant operations, i.e., sum, mean, or max. After $K$ steps, we obtain the final node representations $\boldsymbol{h}_i^{(K)} \forall i \in \mathcal{V}$. These representations are used for tasks like graph-level prediction, employing a permutation-invariant readout function $\hat{\boldsymbol{y}} = \mathrm{g}\left( \left\{ \boldsymbol{h}_i^{(K)} : i \in \mathcal{V} \right\} \right)$ that ensures the output is node order independent.

## 3   RELATED WORK

GNNs extend deep learning models to incorporate graph-structured data Bronstein et al. (2016). Numerous architecture proposals tailored for node, link, and graph prediction tasks have been proposed Scarselli et al. (2009); Gilmer et al. (2017); Wu et al. (2019; 2020); Xu et al. (2018); Corso et al. (2020); Chen et al. (2020). However, these approaches, rely on the complete graph data and often disregard interpretability, which is the focus of our work. More recently, there are works that aim for sparsification Zheng et al. (2020); Hasanzadeh et al. (2020); Rong et al. (2019); Oono & Suzuki (2019); Loukas (2019); Li et al. (2020); Luo et al. (2021); Wickman et al. (2021); Wang et al. (2021) but do not offer control over the interpretability-performance trade-off, are tailored to a specific mode of removal, and focus on tasks other than graph prediction.

Here, we explore relevant sparsification studies that specifically address supervised problems, as they are the main topic of our paper. We categorize these works into two categories: those employing soft removal of information, wherein the sparse graph retains information of the complete graph; and those utilizing hard removal, which closely aligns with our approach.

**Soft removal of information.**   Several architectures Javaloy et al. (2022); Velickovic et al. (2017); Brody et al. (2021) assign varying importance to network edges, simulating soft edge removal whilst maintaining information flow across nodes. DiffPool Ying et al. (2018) hierarchically removes parts of the original graph, distilling the graph's representation into a single vector used for prediction. TopK Gao & Ji (2019), uses a GNN to update node features and subsequently selects the top $k$ most relevant nodes based on their features. It is worth noting that even though part of the network is dropped, the remaining subgraph contains information from the entire network. In contrast, in our framework, the graph predictor does not have access to the dropped information.

**Hard removal of information.**   TopK Gao & Ji (2019) can also be used to select the $k$ nodes based solely on their own information. However, the number of nodes is fixed and cannot dynamically change for different graphs. SUGAR Sun et al. (2021) takes a different approach by identifying discriminative subgraphs. It introduces a reinforcement pooling module trained with Q-learning Mnih et al. (2015) to adaptively select a pooling ratio when recombining subgraphs for classification. Unlike our work, SUGAR does not prioritize sparsity and focuses on finding the most discriminative subgraph that maximizes performance. In contrast, our approach takes a different perspective, as we design the reinforcement learning framework to provide a way to control the trade-off between interpretability and performance. We utilize policy gradient methods to capture uncertainty in the sparsity process and allow for different modes of removal.

Finally, there is extensive literature that aims to understand which subgraph of the input graph is most relevant for making predictions in GNNs Ying et al. (2019); Luo et al. (2020); Yuan et al. (2020); Numeroso & Bacciu (2021); Yuan et al. (2021); Yu & Gao (2022); Schlichtkrull et al. (2021). However, these methods focus on explaining a GNN predictor trained using the original graphs, while our goal is to obtain a GNN that already utilizes a minimal part of the original subgraph. These methods could be used in a subsequent step to further interpret the predictions.

## 4   INTRODUCING THE GCIP FRAMEWORK

In this section, we present a novel framework designed to achieve interpretable predictions in graphs by effectively controlling the trade-off between interpretability and performance, which we refer to as GCIP. Our framework consists of various components, each carefully designed to address this objective. Firstly, we introduce the bi-level optimization procedure, which has been utilized in prior research Sun et al. (2021); Wickman et al. (2021), and enables the simultaneous pursuit of sparsity and performance. Next, we provide a detailed explanation of the sparsity seeker component, which is based on reinforcement learning (RL) and constitutes one of the main contributions of this paper. In contrast to previous work Sun et al. (2021); Wickman et al. (2021), our approach allows for fine-grained control over the trade-off between performance and sparsity at the node/edge removal level.

### 4.1 OPTIMIZING FOR SPARSITY AND PERFORMANCE

Performing graph classification tasks with minimal input graphs poses a challenge that involves two interconnected objectives: maximizing performance and maximizing sparsity. To address this challenge, we propose a two-level iterative optimization approach inspired by previous work Sun et al. (2021); Wickman et al. (2021):

$$\phi^* = \arg\min_{\phi} \mathcal{L}_{\text{spa}} \left( \theta^\star(\phi), \phi, \mathcal{D}^{val} \right) \tag{3}$$

$$\text{s.t. } \theta^\star(\phi) = \arg\min_{\theta} \mathcal{L}_{\text{perf}} \left( \theta, \phi, \mathcal{D}^{tr} \right) \tag{4}$$

This optimization procedure aims to simultaneously maximize predictive performance (Equation (4)) and ensure graph sparsity (Equation (3)). The nested structure of the problem implies that achieving an optimally sparse graph requires a high-performing predictive model. We employ gradient descent for both optimizations, with a larger learning rate for the inner optimization Zheng et al. (2021). Appendix B presents an algoirthm summarizing the training procedure.

**Sparsity optimization Equation (3).** This objective is particularly complex due to the combinatorial nature of node/edge removal possibilities, i.e., $v \in \mathcal{V}$ and $(u, v) \in \mathcal{E}$. Inspired by SparRL Wickman et al. (2021), we formulate the sparsification task as a Markov Decision Process (MDP) and address it using graph reinforcement learning Nie et al. (2022), namely parameterizing $\pi$ using GNNs. In contrast to previous work that uses value-based methods, such as Deep Q-Learning Mnih et al. (2015), we employ the policy gradient method Proximal Policy Optimization (PPO) Schulman et al. (2017a) to capture the inherent uncertainty in the sparsification process. We denote the objective of the sparsity optimization by $\mathcal{L}_{\text{spa}}$, which corresponds to maximizing Equation (1). The output of this task is a policy $\pi_\phi$ that allows getting a sparser graph $\mathcal{G}_s$, which is used as input for the graph classification task, i.e. the solution to the outer optimization problem is thus a policy, $\pi_\phi : \mathcal{G} \mapsto \mathcal{G}_s$, where $\mathcal{G}_s$ is a subgraph of $\mathcal{G}$.

**Performance optimization Equation (4).** This objective corresponds to a standard graph-supervised problem, where the objective is to minimize a loss function $\mathcal{L}_{\text{perf}}$, e.g., the cross-entropy loss. Thus the goal is to learn a function $f_\theta : \mathcal{G}_s \to \hat{\boldsymbol{y}}$ with parameters $\theta$ that minimizes the prediction loss.

### 4.2 CONTROLLING THE INTERPRETABILITY-SPARSITY TRADE-OFF

In this subsection, we provide a detailed description of the components of the PPO instantiation within GCIP. This instantiation gives practitioners control over the trade-off between sparsity and performance. In traditional RL problems, an agent takes a sequence of actions over multiple time steps, and the reward it receives at any time step may depend on all its previous actions. This "delayed reward" problem Sutton (1992) is one of the main challenges in RL.

Our formulation considers a simplified and well-known scenario known as the Multi-Armed Bandit Problem Kuleshov & Precup (2014). In this problem, each agent's decision or action is independent, and the reward is immediate. Then, our goal is to find the sparser graph in a single step rather than considering a trajectory as in traditional RL. As a result, our objective simplifies finding a strategy that maximizes the total reward over the graphs in our dataset. As we empirically demonstrate in our experiments in Section 5, this simplified scenario can achieve a good sparsity-performance trade-off. Henceforth, we only need to describe two components: the policy that controls the type of sparsity (on the number of nodes or edges) and the design of the reward function that allows us to control the sparsity level.

#### 4.2.1 POLICY FORMULATION FOR GRAPH SPARSIFICATION VIA NODE OR EDGE REMOVAL

In real-world applications, we typically determine the necessity of creating a sparser input graph by removing nodes or edges a priori, as it depends on the specific use-case. Accordingly, we propose a policy $\pi(\boldsymbol{a}|\mathcal{G}; \phi) = \prod_i \text{Ber}(a_i|\mathcal{G}; \phi)$ modeled with a GNN with parameters $\phi$, which can operate in both modes of removal, i.e., removing nodes or edges.

This policy accepts as input graph $\mathcal{G}$ and outputs an action $\boldsymbol{a}$, in which each element indicates whether to retain ($a_i = 0$) or eliminate ($a_i = 1$) a specific node or edge in $\mathcal{V}$ and $\mathcal{E}$, respectively. Hence, the

primary role of the policy $\pi$ is to orchestrate the decision-making process for information retention or removal. Recognizing that real-world scenarios are seldom binary, the policy is constructed as a distribution to encapsulate the inherent uncertainty. Depending on the objective of either node or edge removal, we define two operational modes:

- **Node removal policy**: $\pi_n\left(\boldsymbol{a}|\mathcal{G};\phi\right)$, where $\boldsymbol{a} \in \{0,1\}^{|\mathcal{V}|}$. In this scenario, the sparser graph $\mathcal{G}_s$ that we use as input for the downstream task has $\mathcal{V}_s = \{v \in \mathcal{V}|a_v = 0\}$ and $\mathcal{E}_s = \{(u,v) \in \mathcal{E}|a_u = 0 \wedge a_v = 0\}$

- **Edge removal policy**: $\pi_e\left(\boldsymbol{a}|\mathcal{G};\phi\right)$ where $\boldsymbol{a} \in \{0,1\}^{|\mathcal{E}|}$. Here, the sparser graph $\mathcal{G}_s$ that we use as input for the downstream task has $\mathcal{V}_s = \mathcal{V}$ and $\mathcal{E}_s = \{(u,v) \in \mathcal{E}|a_v = 0 \wedge a_v = 0\}$.

Two key observations about the above policies. First, the removal of a node implicitly leads to the exclusion of all edges connected to it. Second, the edge removal policy, in contrast, preserves all nodes, thereby facilitating the potential use of this policy for node classification tasks, which is a direction for future work. We would like to highlight that in either case, finding $\mathcal{G}_s$ is an NP-hard problem for both modes of removal given that the number of possible subgraphs grows exponentially with the number of nodes/edges in the original graph, i.e., the number of possible actions for $\pi_n$ is $\sum_{i=1}^{|\mathcal{V}|-1} \binom{|\mathcal{V}|}{i}$ and for $\pi_e$ is $\sum_{i=1}^{|\mathcal{E}|} \binom{|\mathcal{E}|}{i}$.

### 4.2.2 REWARD FORMULATION

The reward function $\mathrm{R}$ fundamentally shapes the policy's behavior. In numerous real-world graph classification tasks Jiang & Luo (2021); Guo et al. (2021), discerning the extent of the graph's information that is useful to the classification a priori is hard and costly to achieve, e.g., manual labeling Buhrmester et al. (2011). However, it is easy for practitioners to target a minimum sparsity level, enhancing interpretability. To reflect such intent, we design a reward function as a linear combination of a component that seeks performance $\mathrm{R}_p$ and a component that encourages sparsity $\mathrm{R}_s$:

$$\mathrm{R} = \begin{cases} \lambda\mathrm{R}_p(\hat{\boldsymbol{y}}_s) + (1-\lambda)\mathrm{R}_s(\mathcal{G}_s) & \text{if } \hat{\boldsymbol{y}}_s = \boldsymbol{y}, \\ -\lambda\mathrm{R}_p(\hat{\boldsymbol{y}}_s) - (1-\lambda)\mathrm{R}_s(\mathcal{G}_s) & \text{if } \hat{\boldsymbol{y}}_s \neq \boldsymbol{y} \wedge \hat{\boldsymbol{y}} = \boldsymbol{y}, \\ 0 & \text{if } \hat{\boldsymbol{y}}_s \neq \boldsymbol{y} \wedge \hat{\boldsymbol{y}} \neq \boldsymbol{y}. \end{cases} \tag{5}$$

The parameter $\lambda \in [0,1]$ empowers the practitioner to emphasize either performance ($\lambda = 1$) or sparsity ($\lambda = 0$). This reward value differs depending on the prediction of the sparse graph $\hat{\boldsymbol{y}}_s = \mathrm{f}_\theta(\mathcal{G}_s)$ and the original graph $\hat{\boldsymbol{y}} = \mathrm{f}_\theta(\mathcal{G})$. The top scenario corresponds to $\hat{\boldsymbol{y}}_s = \boldsymbol{y}$ when the prediction based on the subgraph $\mathcal{G}_s$ is correct. In this case, we offer a positive reward $\mathrm{R} \in [0,1]$. The middle scenario refers to instances where employing a subgraph $\mathcal{G}_s$ as the classifier's input changes the prediction from correct to incorrect. As this behavior is undesirable, we assign a negative reward $\mathrm{R} \in [-1,0]$. Intuitively, we aim to increase uncertainty in the prediction and discourage removal. The bottom scenario arises when both the original and sparse graph predictions are incorrect, a middle-ground situation that yields a neutral reward $\mathrm{R} = 0$. In the following, we describe how we design both components to achieve the desired trade-off of sparsity and performance.

**Performance Reward.** Our goal is to find policies that enhance performance. To do so, we can employ the cross-entropy loss, usually applied in classification tasks. Yet, this approach is undesirable due to its unbounded nature, making it non-comparable with the sparsity component of the reward. More interestingly, performance can be improved via the entropy of the predictions: $\mathrm{H}(\hat{\boldsymbol{y}}) = -\sum_{k=1}^{K} \hat{y}_k \log \hat{y}_k$. Higher entropy indicates higher uncertainty and vice versa. Given that entropy is a bounded metric for a discrete variable, with $\mathrm{H}_{\max}(\hat{\boldsymbol{y}}) = \log K$, we propose its inclusion in the reward function in a normalized fashion, as $\mathrm{R}_p(\hat{\boldsymbol{y}}) = 1 - \frac{\mathrm{H}(\hat{\boldsymbol{y}})}{\mathrm{H}_{\max}(\hat{\boldsymbol{y}})}$ such that $\mathrm{R}_p \in [0,1]$. In Equation (5), we can observe this component appears as a negative or positive term depending on whether we want to reward (reduce entropy, thus improve performance) or penalize behavior (introduce entropy, trying to change prediction thus improve performance.)

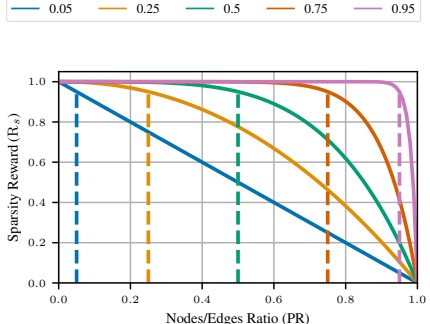

Figure 2: The plots above depict the evolution of different metrics over training epochs, with the shaded area indicating the standard deviation across all validation sets. In the Node Ratio figure we can observe a horizontal line at 1.0 for the model trained on edge policy (GCIP$_E$) as we use all nodes for predicting, even if all the edges of a node have been removed.

**Sparsity Reward.** We also aspire to achieve maximal sparsity, facilitating easier identification of the information for classification prediction. To incentivize this, depending on the policy mode, we propose a reward that penalizes the *nodes ratio* $\mathrm{PR}_n = \frac{|\mathcal{V}_s|}{|\mathcal{V}|}$ or *edges ratio* $\mathrm{PR}_e = \frac{|\mathcal{E}_s|}{|\mathcal{E}|}$ kept,

that is, the proportion of nodes or edges we keep from the original graph. To control the sparsity level, we introduce a parameter $d \in [0, 1]$ representing the *maximum desired nodes/edges ratio* we would like to keep.

Then, we define the reward as $\mathrm{R}_s(\mathcal{G}_s) = 1 - \mathrm{PR}^{\tilde{d}}$ where $\tilde{d}$ is a transformation of $d$ such that $\mathrm{R}_s = 0.95$ when $\mathrm{PR} = d$. The inline figure depicts the variation of $\mathrm{R}_s$ with respect to $\mathrm{PR}$ for distinct $d$ values. Each vertical line represents a unique $d$ value, extending upwards until $\mathrm{R}_s = 0.95$ for all instances. We can observe that to seek sparsity; one should prefer a smaller $d$ value

Figure 1: Evolution of the sparsity reward $\mathrm{R}_s$ over the node/edge ratio for different values of the *maximum desired nodes/edges ratio* $d$.

(e.g., blue or yellow curves). This results in a slower increase in $\mathrm{R}_s$ with node/edge removal, allocating higher rewards only for substantial removal. Conversely, when $d$ approaches 1 (e.g., pink curve), $\mathrm{R}_s$ close to 1 are awarded even for retaining most nodes/edges.

The design of reward functions poses a profound challenge in reinforcement learning, and more sophisticated designs could be employed to tackle tasks beyond classification. In the following section, we present empirical evidence supporting that even this basic choice effectively serves the objective of providing a hyperparameter controlling sparsity and performance.

## 5 EXPERIMENTS

This section presents a thorough ablation study of GCIP and a comparison with baselines. We structure the section into the following subsections. Firstly, we provide insights into the training procedure of GCIP by analyzing the evolution of rewards, performance, and sparsity. Then, we conduct an ablation study to elucidate the effects of choosing different values of $\lambda$ and different desired ratios $d$ on performance. Finally, we provide a comparison of our model with relevant baselines.

**Proposed approaches.** We evaluate the two information removal modes of the proposed framework, described in Section 4.2.1. We denote the model with the policy that eliminates nodes as GCIP$_N$ and the model with the policy exclusively removing edges as GCIP$_E$.

**Datasets.** In this study we used seven BioInformatics datasets, including MUTAG Debnath et al. (1991) , DD Dobson & Doig (2003), ENZYMES Borgwardt et al. (2005), NCI1 Wale et al. (2006), NCI109 Wale et al. (2006), PTC Toivonen et al. (2003) and PROTEINS Borgwardt et al. (2005).

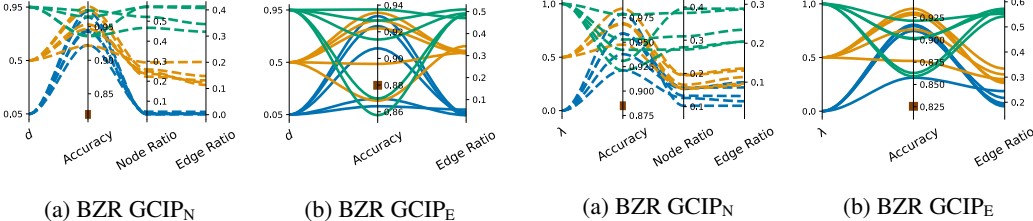

(a) BZR GCIP$_N$    (b) BZR GCIP$_E$    (a) BZR GCIP$_N$    (b) BZR GCIP$_E$

Figure 3: Ablation study on the maximum desired ratio $d$ for 5 different runs. The brown solid rectangle on the accuracy axis represents the GIN (baseline) accuracy.

Figure 4: Ablation study on $\lambda$ (for 5 different runs), which controls the importance given to performance or sparsity. The brown solid rectangle on the accuracy axis represents the baseline GIN (baseline) accuracy.

Additionally, we incorporated two chemical compound datasets( BZR Fey & Lenssen (2019a) and COX2 Fey & Lenssen (2019a)) in our study.

**Experimental setup.** Aiming for computational efficiency, we cross-validate the hyperparameters (e.g., learning rate, number of layers) of the GNN graph-level classifier on GIN. We select the optimal configuration on the validation set and utilize it to train the remaining models. This procedure partially explains GIN's superior performance. We do 5-fold cross-validation to report standard deviation values of the test set on the last epoch. All experiments have been executed on a single CPU with 10GB RAM. For each dataset, we plotted the progress on the validation sets on the above-mentioned four metrics. We use the Torch Geometric Fey & Lenssen (2019b) package for the implementation of the baselines (whenever available), and we provide the implementation of GCIP as well as the necessary scripts to reproduce the experiments at `https://github.com/XXXX/XXXX`. For comprehensive details, refer to Appendix A.

### 5.1 Ablation study

In this section, we provide an ablation study on the proposed framework. We explore training stability and we analyze the effects of key hyperparameters. Furthermore, in Appendix C.2, we show that our approach outperforms a naive baseline based on random node/edge removal. In Appendix C.3, we confirm the reproducibility of results with various base GNN architectures.

**Training** Figure 2 shows the evolution of accuracies, PPO rewards, node ratio, and edge ratio on the validation set of several datasets. We can observe all the metrics show a stable evolution over the epochs, and they converge by the $800^{th}$ epoch, which indicates the bi-level optimization is working. Results for the remaining dataset can be found in Appendix A.

**Analysis of the impact of $\lambda$ and $d$.** The reward function, presented in Equation (5), leverages two hyperparameters—$\lambda$ and the maximum desired nodes/edges ratio $d$—to balance performance and sparsity. The effect of these parameters is illustrated for both GCIP$_N$ (dashed lines) and GCIP$_E$ (solid lines) in Figure 3 and Figure 4. Each line denotes a distinct run, with the solid brown rectangle showing the GIN (baseline) accuracy. Refer to Appendix C for corresponding analyses on the rest of the datasets. Our approach's design efficacy is confirmed by Figure 3, which demonstrates $d$'s impact across its range $\{0.05, 0.5, 0.95\}$. Lower $d$ values induce greater node/edge deletion and sparser graphs. For instance, at $d = 0.05$ (blue line), only around 5% of original nodes/edges remain, indicating that $d$ provides an intuitive mechanism for manipulating solution sparsity. Figure 4 conveys the influence of $\lambda \in \{0.0, 0.5, 1.0\}$, the parameter responsible for prioritizing sparsity or performance. Here, we also achieve sparsity with lower values of $\lambda$ (see blue line), albeit less extreme than with $d$. Even for $\lambda = 0.0$, around $15 - 20\%$ of nodes and edges persist, as seen in Figure 4a and Figure 4b, respectively. As depicted in Figure 3a and Figure 4b, lower $d$ and $\lambda$ values (blue lines) often correspond to decreased accuracy. Conversely, higher values (green line) tend to promote accuracy, although fold variability complicates drawing conclusions. In light of these findings, we recommend practitioners to utilize $d$ to fine-tune sparsity levels and opt for $\lambda = 1.0$ when performance is the primary objective and sparsity is inconsequential.

Table 1: Test set accuracy results. We show the mean over 5 independent runs and the standard deviation as the subindex. The last row includes the average ranking of the model across datasets. Best performing models on average are indicated in bold.

| Dataset | Full Models | | | Sparse Models | | | |
|---|---|---|---|---|---|---|---|
| | GIN | TopK$_{soft}$ | DiffPool | SUGAR | TopK$_{hard}$ | GCIP$_N$ | GCIP$_E$ |
| BZR | 81.95 $_{2.78}$ | 79.51 $_{3.70}$ | 70.24 $_{5.95}$ | - | 76.10 $_{6.31}$ | 79.90 $_{4.29}$ | **82.39** $_{5.07}$ |
| COX2 | **84.58** $_{4.56}$ | 80.42 $_{5.02}$ | 65.83 $_{6.65}$ | - | 71.67 $_{12.02}$ | 75.49 $_{6.33}$ | 83.59 $_{2.82}$ |
| DD | 73.11 $_{2.45}$ | 76.64 $_{4.54}$ | - | - | **77.65** $_{2.42}$ | 75.51 $_{1.86}$ | 74.40 $_{2.12}$ |
| ENZYMES | **72.33** $_{4.35}$ | 68.67 $_{6.50}$ | 31.00 $_{6.66}$ | 16.67 $_{23.57}$ | 71.67 $_{4.86}$ | 63.70 $_{3.89}$ | 49.53 $_{5.29}$ |
| MUTAG | **86.00** $_{4.18}$ | 82.00 $_{6.75}$ | 69.00 $_{12.21}$ | 76.34 $_{3.80}$ | 76.00 $_{4.18}$ | 74.63 $_{2.12}$ | 74.32 $_{6.34}$ |
| NCI1 | **82.04** $_{1.26}$ | 80.78 $_{0.24}$ | 69.93 $_{3.24}$ | 49.95 $_{35.58}$ | 74.37 $_{1.62}$ | 73.66 $_{2.74}$ | 73.51 $_{1.02}$ |
| NCI109 | **81.59** $_{1.91}$ | 78.26 $_{1.02}$ | 67.49 $_{2.60}$ | 49.65 $_{35.71}$ | 74.09 $_{5.99}$ | 71.88 $_{3.94}$ | 72.48 $_{2.78}$ |
| PROTEINS | 72.86 $_{4.22}$ | 71.96 $_{5.74}$ | 66.42 $_{7.19}$ | 59.57 $_{43.01}$ | 72.14 $_{3.54}$ | 73.49 $_{7.27}$ | **73.50** $_{5.03}$ |
| PTC | 59.44 $_{8.91}$ | 56.75 $_{7.32}$ | 54.44 $_{11.10}$ | 56.14 $_{8.95}$ | 61.11 $_{8.56}$ | 52.92 $_{8.67}$ | **66.06** $_{4.31}$ |
| **Accuracy rank** | **2.0** | 3.0 | 6.12 | 6.0 | 3.22 | 4.11 | 3.22 |

## 5.2 PERFORMANCE COMPARISON

In this section, we thoroughly evaluate GCIP by comparing our approach to baselines and competing methods on nine datasets. We assess our model's performance over the test set on three metrics: accuracy and node and (edge) ratio measured in % of nodes (edges) kept in the graph). Reducing the node and edge ratio percentages increases graph sparsity, making the models more interpretable.

**Baselines.** We divided the baselines into two categories: *Full* model and *Sparse* model approaches. As *Full* model approaches, which use the full graph information, we compare with GIN Xu et al. (2018) (as the vanilla approach), DiffPool Ying et al. (2018), and TopK$_{soft}$—an implementation of TopK Cangea et al. (2018) that utilizes node embeddings coming from a GNN to select the top $k$ nodes, thus incorporating global information. As *Sparse* models, we compare with SUGAR Sun et al. (2021) [1] and TopK$_{hard}$—the implementation of TopK Cangea et al. (2018) where the node embeddings used to select the top $k$ comes from a multi-layer perceptron, hence the sparse graph does not contain global information.

**Accuracy results.** We use test accuracy to compare all models on the nine datasets. Table 1 summarizes the results. It can be observed that although GIN has the best accuracy on almost all the datasets, it is worth noting that GCIP gives competitive performance. Among the *Sparse* models, we observe that SUGAR underperforms the other models and, remarkably, GCIP$_E$ achieves the best ranking (shared with TopK$_{hard}$) while using a considerably sparser representation, as illustrated in Table 2.

**Interpretability results.** We perform a node and edge ratio analysis on the three best-performing *Sparse* models in terms of accuracy: TopK$_{hard}$, GCIP$_N$, and GCIP$_E$. The results are summarized in Table 2. It is important to note that GCIP$_E$ consistently maintains a node ratio of 100%, owing to its edge removal strategy that preserves all nodes irrespective of their connectivity. Among the models, GCIP$_N$ consistently exhibits the highest rank in both node and edge ratios, indicating that it eliminates a larger number of nodes and edges from the original graph. Consequently, the graph classifier operates on a reduced, and thus more interpretable, subgraph. This might account for the slight performance reduction observed for GCIP$_N$ in Table 1 compared to GCIP$_E$. Ultimately, the selection between these models depends on the specific objectives of the practitioner.

**Time complexity.** Training and inference are significantly faster in methods using a simple forward pass than in RL-based methods like SUGAR and GCIP. Notably, our observations reveal that GCIP is at least as fast as SUGAR in training, and up to 10 times faster in inference. Refer to Appendix D for complete quantitative results on all datasets.

---

[1] We use the official implementation of SUGAR to extract the results on the above datasets. We believe our evaluation of a held-out test set may explain the disparity between the results we report and those in Sun et al. (2021). Instead, the available implementation of SUGAR evaluates on the same validation set used to select the best model.

Table 2: Node/Edge ratio (shown in %) for the graphs in the test set for *Sparse* models on 9 different datasets. Numbers in parentheses indicate the ranking of the model for each dataset. The last two rows indicate the average ranking of the models across datasets in terms of node and edge sparsity, respectively. Best performing models on average are indicated in bold.

| Dataset | Nodes/Edges % | Sparse Models | | |
| --- | --- | --- | --- | --- |
| | | TopK$_{hard}$ | GCIP$_N$ | GCIP$_E$ |
| BZR | Node Ratio (%) | $31.28 \pm 0.04$ (2) | $\mathbf{18.87 \pm 1.67}$ **(1)** | $100.0 \pm 0.0$ |
| | Edge Ratio (%) | $22.09 \pm 1.86$ (2) | $\mathbf{12.80 \pm 1.37}$ **(1)** | $29.07 \pm 2.89$ (3) |
| COX2 | Node Ratio (%) | $50.63 \pm 0.10$ (2) | $\mathbf{17.48 \pm 1.72}$ **(1)** | $100.0 \pm 0.0$ |
| | Edge Ratio (%) | $44.50 \pm 3.68$ (3) | $\mathbf{11.52 \pm 2.82}$ **(1)** | $20.41 \pm 1.33$ (2) |
| DD | Node Ratio (%) | $90.25 \pm 0.02$ (2) | $\mathbf{66.64 \pm 2.73}$ **(1)** | $100.0 \pm 0.0$ |
| | Edge Ratio (%) | $81.19 \pm 0.34$ (3) | $\mathbf{44.20 \pm 3.40}$ **(1)** | $58.14 \pm 0.77$ (2) |
| ENZYMES | Node Ratio (%) | $91.74 \pm 0.08$ (2) | $\mathbf{81.84 \pm 1.76}$ **(1)** | $100.0 \pm 0.0$ |
| | Edge Ratio (%) | $84.25 \pm 0.63$ (3) | $67.16 \pm 2.88$ (2) | $\mathbf{47.16 \pm 12.87}$ **(1)** |
| MUTAG | Node Ratio (%) | $73.04 \pm 0.89$ (2) | $\mathbf{66.00 \pm 9.87}$ **(1)** | $100.0 \pm 0.0$ |
| | Edge Ratio (%) | $\mathbf{55.00 \pm 1.65}$ **(1)** | $62.15 \pm 14.50$ (2) | $95.42 \pm 6.27$ (3) |
| NCI1 | Node Ratio (%) | $71.78 \pm 0.05$ (2) | $\mathbf{32.08 \pm 2.77}$ **(1)** | $100.0 \pm 0.0$ |
| | Edge Ratio (%) | $48.82 \pm 0.52$ (3) | $\mathbf{19.89 \pm 5.17}$ **(1)** | $25.69 \pm 2.02$ (2) |
| NCI109 | Node Ratio (%) | $91.74 \pm 0.06$ (2) | $\mathbf{53.88 \pm 2.06}$ **(1)** | $100.0 \pm 0.0$ |
| | Edge Ratio (%) | $89.27 \pm 0.33$ (3) | $41.71 \pm 4.09$ (2) | $\mathbf{40.65 \pm 5.76}$ **(1)** |
| PROTEINS | Node Ratio (%) | $92.27 \pm 0.24$ (2) | $\mathbf{85.69 \pm 14.06}$ **(1)** | $100.0 \pm 0.0$ |
| | Edge Ratio (%) | $85.76 \pm 0.41$ (3) | $75.63 \pm 22.91$ (2) | $\mathbf{47.60 \pm 8.10}$ **(1)** |
| PTC | Node Ratio (%) | $94.90 \pm 1.08$ (2) | $\mathbf{39.16 \pm 7.52}$ **(1)** | $100.0 \pm 0.0$ |
| | Edge Ratio (%) | $89.94 \pm 1.95$ (3) | $\mathbf{18.06 \pm 8.06}$ **(1)** | $52.30 \pm 1.68$ (2) |
| **Average Rank** | Node Rank | 2.00 | **1.00** | - |
| | Edge Rank | 2.67 | **1.44** | 1.89 |

## 6 CONCLUSIONS

Graph Neural Networks (GNNs) excel in graph-level tasks but suffer from interpretability issues due to their complexity. Past endeavors to enhance interpretability, focusing on post-hoc explanations or *sparsity* during training, were limited by reliance on complete graph information and lack of control over the performance-sparsity trade-off.

Addressing these, we proposed GCIP, a novel framework that relies on a bi-level optimization to target both performance and sparsity. The latter is achieved by employing reinforcement learning and offers direct control over the performance-interpretability trade-off via two hyperparameters, allowing to select two modes of information removal: at node or edge level. Our empirical evaluation conducted on nine different graph classification datasets provides evidence that our approach not only competes in performance with the baselines that use the entire graph information, but also relies on significantly sparser subgraphs. The resulting GNN-based predictions, therefore, are more interpretable, addressing the primary motivation of our work.

The main limitation of the proposed approach lies in the increased training and inference time introduced by the reinforcement learning component. Therefore, future research could focus on enhancing its efficiency. Other interesting future works could be applying GCIP$_E$ to node classification tasks and the modification of the current reward function to penalize specific types of errors. Similarly, adapting the reinforcement learning pipeline for metric learning scenarios could further boost performance.

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
