# OpenReview forum: "Prediction Tasks in Graphs: a Framework to Control the Interpretability-Performance Trade-off"
_ICLR.cc/2024/Conference — Submitted to ICLR 2024_

### Official Review · Reviewer_CD1o · 2023-10-26

**Soundness:** 3 good
**Presentation:** 3 good
**Contribution:** 3 good
**Rating:** 6
**Confidence:** 2

**Summary:**

GNNs are hard to interpret. The authors aim to produce a method which only relies on small sparse subsets of the graph for improved interpretability (and therefore utility to practitioners). Furthermore, they aim for the user to manually be able to tune the "interpretability vs performance tradeoff".

The ML task is graph classification and the approach is based off of reinforcement learning.

**Strengths:**

Interpretability is a challenging and important problem and it is not very well explored in the GNN domain. The method presented seems to be a natural step forward building on previous work towards this problem that gives the user the ability to flexibly balance important tradeoffs.

The numerical results are generally strong.

**Weaknesses:**

This paper was quite hard to read for someone (me) with minimal knowledge of RL. This problem was compounded by the fact that several terms were used in e.g., equation (1) without being well defined. That said, I recognize these issues are somewhat inevitable due to the fact that the paper draws from multiple different subareas of ML and the page-limited format of a conference submission.

The results are generally strong but somewhat inconsistent. It would be good to have better theoretical insights as to when the proposed method is / is not appropriate.

Minor Issues (unimportant for accept/reject but should be fixed in camera ready):

In the notation it should be $V=\{1,\ldots,n\}$ not $V\in 1,\dots,n$. Additionally, the curly braces are missing from $y_i\in \{1,\ldots,K\}$.

Several of the terms in Equation (1) are used without being defined, e.g. ``clip" and $a_t$.

Typo in line four: Missing space before ``However"

In the references, "euclidean" should have a capital E. Please check for similar issues throughout.

**Questions:**

Why is the method particularly ineffective on Enzymes?

Do you have any intuition on when the method (or other sparsity methods) will be appropriate?

Is it possible to adapt this method to other graph ML tasks, e.g., node classification, graph regression, etc?

---

### Official Review · Reviewer_Zge8 · 2023-11-01

**Soundness:** 3 good
**Presentation:** 3 good
**Contribution:** 2 fair
**Rating:** 5
**Confidence:** 4

**Summary:**

The paper seeks to improve GNN interpretability by sparsification of graphs during training. A bilevel optimization framework is adopted where the outer loop controls sparsity and the inner loop seeks to maximize classification performance. An RL paradigm is adopted in the  outer loop. The trade-off between sparsity and performance is controlled by a hyperparameter. The overall thesis is that a sparser graph should be easier to interpret. Empirical results on nine different classification benchmarks from the chemical domain show that that the method competes in performance with baselines that use information from the whole graph, while relying on sparser subgraphs.

**Strengths:**

1. The formulation of the problem as a bilevel optimization problem and the different loss functions is nice and the problem is solved elegantly and efficiently.
2. The variants of pruning nodes and edges are meaningful.
3. The method works well on the different benchmarks as compared to the competing baselines.

**Weaknesses:**

1. There is a correlation between the sparsity and explainability but how strong is that correlation? Can that be quantified or measured? Are the authors able to show that the sparser graphs that they find are easier to explain?
2. There is a wide variability in the level of sparsification achieved on the different datasets? Can the authors explain that? By the way, I did not see the details on the sizes of the datasets; may be I missed that.
3. Could the authors use a spectral sparsification benchmark? https://arxiv.org/abs/0808.4134 is one such reference but there are others. As other baselines, there is gpool (https://arxiv.org/abs/1905.05178), Eigenpooling (https://arxiv.org/abs/1904.13107), and self-attention pooling (https://arxiv.org/pdf/1904.08082.pdf). It would be good to compare against them too. There may other more recent methods.
4. Is there a constraint that the graph remains connected during pruning?

**Questions:**

Please see the weaknesses above.

---

### Official Review · Reviewer_X2ML · 2023-11-02

**Soundness:** 3 good
**Presentation:** 2 fair
**Contribution:** 2 fair
**Rating:** 5
**Confidence:** 4

**Summary:**

Graph neural networks (GNNs) solve graph-level tasks in diverse domains. In this paper, the authors aim to minimize the size of an input graph while maintaining the performance. The authors formulate GNN training as a bi-level optimization task, where the trade-off between interpretability and performance can be controlled by a hyperparameter. The proposed framework relies on reinforcement learning.

**Strengths:**

1. A real graph may contain many redundant and even noisy edges. Preserving the performance of a GNN with increased sparsity seems like a practical problem.
2. Reinforcement learning seems like a reasonable approach to optimize the sparsity of a graph, which is discrete and hard to optimize through gradient descent.
3. The authors provide the full algorithm and detailed experimental results in the appendix, which helps a reader to get a better understanding of the paper.

**Weaknesses:**

1. The authors claim that the interpretability of a GNN can be improved by sparsifying an input graph. However, that connection seems unclear to me, since a GNN can still be difficult to interpret even on a sparse graph. The sparsity level achieved by the proposed method is between 10 to 85%, which does not seem to bring dramatic improvement of interpretability.
2. The authors present the accuracy and sparsity at different tables, making it hard to evaluate the performance of the proposed method. I suggest drawing a scatter plot with one axis for accuracy and the other for sparsity.
3. Only one base model, GIN, is used in experiments. As the authors propose a general framework for balancing the performance and performance of a GNN, it would be better to include at least two different base models and show the success of the approach.
4. Table 8 in the appendix shows that the proposed approach is up to 100x slower than the base model. That means one can test the base model 100 times with different sparsity levels. Can the authors show that their approach is better than this “random guess” given the same time budget?

**Questions:**

1. Why does “the nested structure of the problem imply that achieving an optimally sparse graph requires a high-performing predictive model” in Section 4.1?
2. There is a typo in the Edge Removal Policy in Section 4.2.1.

---

### Official Review · Reviewer_CKsQ · 2023-11-10

**Soundness:** 3 good
**Presentation:** 3 good
**Contribution:** 2 fair
**Rating:** 3
**Confidence:** 3

**Summary:**

The paper formulates GNN training as a bi-level optimization task that achieves the trade-off between interpretability and performance. The authors do more experiments to show the results are better than the baseline methods.

**Strengths:**

1 It is novel to propose the bi-level optimization to achieve the trade-off between the interpretability and performance.
2. Use the reinforcement learning to iteratively maximize predictive performance and sparsity by removing edges or nodes from the input graph.

**Weaknesses:**

1 The authors also need the metric to measure interpretability. The authors only show the accuracy of the model after training the sparsity graph. There are no experiments to show the faithfulness of the interpretability after training. The author can use the metric of interpretability, such as Fidelity.
2. No comparison with interpretability methods. For the interpretability methods, such as gnnexplainer, pgexplainer, they are also explain graph classification. The authors can select top-k nodes according to the mask matrix, and compare the results with the authors’ method.
3. The authors should describe the graph dataset, such as the nodes and edges in the dataset. I'm not sure whether the authors tested on large graph datasets. Is the proposed method effective on large graph datasets? I think it is hard to apply to large datasets because the upper-level optimization problem is trained using reinforcement learning, this will take more time. If it cannot used in large datasets, this will lead to confusion about the motives. If the graph data is relatively small, traditional interpretability methods can achieve a better trade-off between sparsity and accuracy.
4. Comparing the runtime against other baseline method/s will be helpful in identifying the runtime performance tradeoff.
5. The authors train GNN model as a bi-level optimization task, what is the performance of using multi-objective optimization method to train the model? The authors should add some multi-objective optimization baselines and compare the results. For a simple example, considering the accuracy and the interpretability, you can define the loss function L_{C}(for accuracy) and L_{sparsity}(for interpretability) , and the total loss function L=L_{C} +L_{sparsity}, you can use this loss function to train the model.
6 If train the GNN model first, and then fixed model parameters, then train the reinforcement learning to ensure the sparsity, what’s the performance of this method? And what’s the difference between this method and the bi-level optimization? I'm a little confused about the motivation for the bi-level optimization.
7 Some places in the paper are confusing. For example, in equation (3) and equation (4), what’s the meaning of the \theta and \phi? It is best to explain the meaning of the symbols below the formula. In page 5, edge removal policy, there are some errors in the definition of the \mathcal{E}_S.

**Questions:**

See the weakness

---

### Meta-Review · Area_Chair_n1z7 · 2023-12-19

**Metareview:**

The reviewers found this paper interesting but have concerns mainly centered on the empirical study, from more quantitative evaluations on the interpretability side to more benchmark tasks. The overall recommendation leans toward rejection without these issues fixed.

**Justification For Why Not Higher Score:**

Require improvements on the empirical study, from more quantitative evaluations on the interpretability side to more benchmark tasks.

**Justification For Why Not Lower Score:**

N/A

---

### Decision · Program_Chairs · 2024-01-16

Reject